# Dark Personality Traits and Online Behaviors: Portuguese Versions of Cyberstalking, Online Harassment, Flaming and Trolling Scales

**DOI:** 10.3390/ijerph20126136

**Published:** 2023-06-15

**Authors:** Ângela Leite, Susana Cardoso, Ana Paula Monteiro

**Affiliations:** 1Centre for Philosophical and Humanistic Studies (CEFH), Universidade Católica Portuguesa, 4710-362 Braga, Portugal; 2Research Center in Sports Sciences and Human Development, CIDESD, Universidade de Trás-os-Montes e Alto Douro, 5000-801 Vila Real, Portugal; susana.cardoso@utad.pt; 3Department of Social Sciences and Behavior, University of Maia, Av. Carlos Oliveira Campos, 4475-690 Maia, Portugal; 4Departamento de Educação e Psicologia, Universidade de Trás-os-Montes e Alto Douro, 5000-801 Vila Real, Portugal; apmonteiro@utad.pt; 5CIIE—Center for Research and Intervention in Education, University of Porto, 4200-135 Porto, Portugal

**Keywords:** cyberstalking scale, dark triad of personality, online behavior scales, validation

## Abstract

The main objective of this study is to assess moderation effects of online behaviors between personality traits and addiction to Internet. To this end, four instruments were validated for Portuguese version through confirmatory factor analysis and exploratory factor analysis (Study 1) Multiple regression analysis was applied to examine the personality predictors of specific online behaviors while controlling for gender and age; and moderation effects were assessed (Study 2). Results showed good psychometric properties for the four validated scales. Machiavellianism is positively associated with all the dimensions of this study. Psychopathy is positively associated with total Cyberstalking, Cyberstalking Control, Flaming and Trolling. Narcissism is positively associated with all the dimensions, except Online Harassment and Flaming. Machiavellianism is positively associated with Addiction to Internet through Cyberstalking, Flaming and Trolling. Psychopathy is positively associated with Addiction to Internet through Cyberstalking Control and Flaming. Narcissism is also positively associated with Addiction to Internet through Cyberstalking and Trolling. This study demonstrates that dimensions of the dark triad of personality play an important role in Internet addiction through online behaviors. The results of this study have theoretical and practical implications: on the one hand, they reinforces the findings of other studies showing that dimensions of the dark personality triad play an important role in Internet and social network addition, contributing to the literature; and, on the other hand, on a practical level, they allow to conduct awareness campaigns in communities, schools, and work to understand that one can be exposed to unpleasant situations due to behaviors that some people with personality traits of Machiavellianism, narcissism and/or psychopathy that may cause problems affecting the mental, emotional and psychological health of others.

## 1. Introduction

It is almost impossible to live nowadays without the Internet and hundreds of online applications [1]. Although certainly beneficial, they can also represent a source of problems when used problematically or additively [1,2,3], and/or the emergence of various forms of abuse [4,5] and unwanted and problematic digital contact [6,7].

The addition to the Internet is an extensive concept that encompasses a multiplicity of behaviors and impulse control problems in its use [3], with characteristics similar to other types of dependence, namely, salience, tolerance, mood modification, conflict, abstinence and relapse [2,8].

Cyberstalking, also known in the literature as electronic or virtual stalking, includes the use of the Internet (or other computerized device), to harassing or chasing other persons, by systematic and undesirable actions, causing suffering to the targets of these behaviors [5]. Cyberstalking can include behaviors such as maintaining remote surveillance, constant contact with and/or direct threats against the victim [9]. In fact, online harassment may become a public health issue because of the Internet’s globalization [6,10]. Similarly, online harassment is considered a form of cyberbullying that consists of repeatedly sending offensive, threatening or intimidating messages to someone via email, short message (or messaging) service (SMS), multimedia messaging service (MMS) or others [11]. This can also arise in the form of offensive sexual messages [12].

Literature has reported a number of harmful consequences for mental health of victims of cyberstalking and/or harassment. The negative psychological impact of the cyberstalking and/or online harassment manifests itself through psychopathological symptoms such as anxiety, depression, posttraumatic stress disorder and panic attacks; through negative emotions such as sadness, anger, fear, shame, embarrassment, isolation and low self-esteem; through physical symptoms such as stomach pains and heart palpitation; and even through inappropriate behaviors such as harmful self-behavior [13,14].

In turn, Trolling can be considered a type of online harassment [10]; this behavior is intended to intentionally deceive, destabilize a discussion, and provoke others creating conflict, and evoking hostility and anguish, which causes pleasure in online offenders [4,10]. Although there is no consensus in the literature regarding the definition of flaming, most studies consider that flaming includes sending hostile, aggressive, intimidating, insulting and offensive messages through uninhibited and sarcastic language [7]. These messages and comments do not contribute to the discussion in question, but instead attempt to hurt another person socially or psychologically and impose power on others [15]. According to Hinduja and Patchin [15], trolling is directed to the subject of discussion, while flaming is directed at another participant or other participants in the discussion. Trolls try to cause maximum disruption, annoyance through the arguments presented on the subject online and their words and actions have no sincere basis. In turn, in flaming, the transmission of what the individual thinks is true or correct is done in an aversive and harmful way [15].

It is possible to classify the above-mentioned behaviors (cyberstalking, online harassment, flaming, trolling) as anti-social behavior and online; according to Moor and Anderson [16], antisocial online behavior is any deviant behavior or the deliberate absence of proper behavior that is committed online and has negative online and or offline consequences to whom it is directed (including self-directed behavior).

It should be noted that online antisocial behavior represents a more complex phenomenon and with more serious consequences because it is practiced online or through any technological devices. Thus, the possible anonymity of the perpetrator, the public potential to become uncontrollable, the constant nature of victimization through the permanent presence of information and communications technology (ICT) in everyday life, the increased difficulty in identifying the perpetrator and the power imbalance created by anonymity are perceived by victims as weakening and strengthening the perpetrators [17,18].

Research has shown that the personality traits that make up dark triad play a crucial role in adding to the Internet and social networks [19,20,21]. These traits also appear in the literature as an attempt to explain the problems related to Internet misconduct/online antisocial behaviors. Manuoğlu and Öner-Özkan [10] highlighted the importance of analyzing the role of the dark triad of personality in online behavior, considering that these personality traits can have a negative impact and with serious consequences on these behaviors.

The dark triad assesses aversive personality traits: Machiavellianism, psychopathy and narcissism [22,23]. These traits share some characteristics in common, such as manipulation, selfishness, insensitivity, lack of empathy and affection [24]. Dark triad has been the subject of several studies in the last decade, which has recently led to its extension being suggested to introduce Sadism, thus becoming the dark tetrad [4]. Such insertion is still not consensual [25], although Bonfá-Araujo et al. [26] found that Sadism is the best predictor of aversive online behaviors. Also, Pineda et al. [27] found that Sadism was the strongest predictor of online sexual victimization perpetration. Thus, Machiavellianism is characterized by cunning, cynicism, selfish ambition, difficulties in terms of empathy, manipulation and strategic exploitation of others [25,28]. Regarding narcissism, the traits associated with it are the feeling of grandiosity, individualism and an unrealistic self-image that reflects an idea of superiority, dominance and entitlement in relation to others. It is also characterized by a lack of empathy and feelings of envy [25,29]. It also presents a generalized and cynical view of the world and people [30]. In turn, psychopathy is characterized by insensitivity, low level of empathy, impulsiveness, imprudence, disdain for others, tendency to tease and deception, and also by loquacity and superficial charm [25,28,31].

Several researches reveal that dark triad’s personality traits seem to be positively associated with the problematic addition to the Internet [32] and with online antisocial behaviors, particularly cyberstalking [9,16], online trolling [10,33,34] and cyberbullying [35]. Machiavellianism has been associated with problematic use of social media, trolling in online games, and online self-promotion. Individuals high in Machiavellian traits use social media or online gaming platforms to engage in manipulative interpersonal behavior or deceptive self-promotion, in part, due to their fear of social rejection. Considering that Machiavellianism is negatively associated with positive humor, it is expected that people with a high level of this trait will end up becoming problematic Internet users, since using the Internet corresponds to a (maladaptive) coping strategy to deal with negative feelings [32,36]. Machiavellian individuals can use Internet games, especially violent games, not only for entertainment, but also to satisfy the need to explore and control others [37]. Narcissism is associated with a great involvement with problematic use of the Internet and social networks [20,32,38,39]. People who have high levels of this trait tend to exhibit online self-promotion behaviors (sometimes deceptively), through selfies, posts and video clips. Self-promotion behaviors and the attempt to present a more popular self than the real ones are risk factors for the development of problematic Internet use [38,39]. In turn, psychopathy is associated with emotional dysregulation and a low level of positive mood. Thus, psychopathic individuals tend to seek online activities as a form of coping, but also to seek in these activities a way to obtain greater sensations. They also tend to engage in antisocial online behaviors such as cyberbullying, trolling, cyberstalking intimate partners, and playing violent games [9,20,32].

Sindermann et al. [37] stated that there are associations, in females, between Machiavellianism and psychopathy, on the one hand, and tendencies toward Internet-shopping disorder, on the other. The authors also found an association between psychopathy, on the one hand, and tendencies toward Internet-pornography-use disorder, on the other. Finally, they showed an association between Machiavellianism, on the one hand, and tendencies toward Internet-communication disorder, on the other. In turn, the associations found between the dark triad traits and unspecified Internet use disorder are similar in males and females.

Smoker and March [9] examined the influence of Dark Tetrad on intimate partner cyberstalking and the results revealed moderate associations with psychopathy, Machiavellianism and Sadism, while narcissism was weakly correlated. Also, high scores in the dark triad of personality are related to high levels of perpetration of cyber dating abuse [40].

In the study by Lopes and Yu [34], 135 participants evaluated two fake Facebook profiles regarding their agreement with trolling comments and with the social acceptance of the fake profiles. The authors found that narcissism was related to a tendency to see oneself as superior to the popular profile, while psychopathy was positively associated with trolling behaviors; in addition, individuals with higher scores on psychopathy were more likely to spy a profile that was considered popular. In turn, the results of the systematic review of the literature conducted by Moor and Anderson [16] suggest that psychopathy is the darkest of traits, based on its ability to predict behaviors of “high severity”, namely cyberaggression and technology facilitated sexual violence.

The main objective of this study is to assess the moderation effects of online behaviors between personality traits and addiction to Internet through two studies. To achieve this objective, specific objectives were outlined: (1) to evaluate the psychometric qualities of the instruments used in this study through (1a) assessment of the adjustment to our sample of instruments previously validated for the Portuguese population (Internet Addiction Test and Dirty Dozen Dark Triad); (1b) and the validation for the Portuguese population of the Cyberstalking scale, Online Harassment scale, Flaming scale and Trolling scale; (2) to establish associations between Internet addiction, Machiavellianism, psychopathy and narcissism, cyberstalking, online harassment, flaming and trolling, on the one hand, and the sociodemographic variables, o the another; (3) to establish associations between personality and online behaviors; and (4) to understand if online behaviors moderates the relation between the dark triad of personality and Internet addiction.

## 2. Materials and Methods

### 2.1. Procedures

All procedures followed the Declaration of Helsinki and later amendments or comparable ethical standards. Also, the Scientific Committee of Universidade Católica Portuguesa approved this study that was conducted from July to September 2022. Participants were recruited through social media and assessed through Google Forms, having been informed about the study’s purpose and assured of confidentiality and anonymity of the data. All participants signed the informed consent. The criteria used to select participants were: to be 18 years old or older, and have Portuguese nationality and residence. Exclusion criteria include being fewer than 18, not understanding the content of the questions asked and not using social media. The translation of the four instruments (Cyberstalking Scale, Online Harassment Scale, Flaming Behaviors Scale and Trolling Behavior Scale) was performed according to the International Test Commission (ITC) guidelines for translating and adapting tests [41] and the translation back-translation procedure [42]. The original versions of the instruments were translated from English to Portuguese by two bilingual translators, one psychologists and another from social sciences field. A third bilingual translator (psychologist) carried out a reconciliation of the two translations. A native English speaker from other religious field performed the reconciled version’s back-translation. The first translator compared the back-translated version with the original English versions to achieve linguistic and cultural equivalence consistency. No differences were found between the back-translated and the original versions. A convenience sample of 15 people over 18 years old, with Portuguese nationality and residence were invited to evaluate the items’ relevance to the scales and cultural context. Original and Portuguese versions of the scales can be found in Appendix A.

### 2.2. Instruments

#### 2.2.1. Sociodemographic Questionnaire

The sociodemographic questionnaire includes questions related to gender (man—0; woman—1), age, years of education and if participants are in a romantic relationship (no—0; yes—1).

#### 2.2.2. Psychological Variables

##### Internet Addiction Test

The Internet Addiction Test (IAT) was conceived by Young [43] to measure the extent of a person’s involvement with the Internet; it classifies addictive behavior as mild, moderate or severe impairment. IAT comprises 20 items, rated on a six-point Likert scale (0—‘does not apply’, 1—‘rarely’, 2—‘occasionally’, 3—‘frequently’, 4—‘often’, and 5—‘always’). The sum of all items allows obtaining a global score [43]. Young [43,44] proposed two different cut-off points criteria: the first one (1998) [44] stated that values between 20 and 39 correspond to the average user; between 40 and 69 correspond to a person that has frequent problems because of Internet usage; and between 70 and 100 correspond to Internet addicts. The second cut-off points criteria [43] proposed that values between 0 and 30 correspond to a normal range; between 31 and 49 correspond to mildly addicted; between 50 and 79 correspond to moderately addicted; and between 80 and 100 correspond to severely addicted. Factor analysis of the IAT by Widyanto and McMurran [45] yielded six factors (salience, excessive use, neglecting work, anticipation, lack of control, and neglecting social life) with all factors showing good internal consistency and concurrent validity. Several studies reported reliability coefficients and internal consistency for the IAT, ranging between α = 0.88 and α = 0.93 [46,47]. Pontes et al. [48] validated the IAT for the Portuguese population and found a uni-factorial model for the IAT and also found that IAT is a valid and reliable instrument for measuring IA among Portuguese young adults as demonstrated by its satisfactory psychometric properties (e.g., α = 0.90). In this study it was also found a value of α = 0.90.

##### Dirty Dozen Dark Triad

Jonason and Webster [49] developed and validated a concise, 12-item measure of the Dark Triad: narcissism, psychopathy, and Machiavellianism because an exponential interest in the dark side of human nature lasts. The authors first created 22 candidate items inspired by the original Dark Triad measure; the four items with the strongest loadings on the primary factor were retained from each of the three Dark Triad measures, being that, together, these 12 items constituted the Dark Triad Dirty Dozen and the three factors emerged [49]. Participants answered the 12 items by agreeing or not (1—‘strongly disagree’, 5—‘strongly agree’). Overall, the Dark Triad Dirty Dozen achieved good internal consistency α = 0.83, for psychopathy α = 0.63, Machiavellianism α = 0.72, and narcissism α = 0.79. Pechorro et al. [50] validated this instrument for the Portuguese population and found a three-factor structure invariant across the genders and adequate psychometric properties (internal consistency, convergent validity, discriminant validity, criterion-related validity, and know-groups validity (boys versus girls). Pechorro et al. [50] reported alpha values for psychopathy α = 0.63, Machiavellianism α = 0.72, and narcissism α = 0.79. In this study, it was found a value of psychopathy α = 0.68, Machiavellianism α = 0.73, and narcissism α = 0.81.

##### Cyberstalking Scale

Silva Santos et al. [51] developed a cyberstalking measure for the Brazilian population. Initially, this scale had 15 items, but after exploratory factor analysis and Cronbach’s alpha, five items were removed, remaining 10 items. These items were answered on a five-point Likert-type scale (from 1—‘totally disagree’ to 5—‘totally agree’). The ten items solution presents a Cronbach alpha value of 0.82. A confirmatory factor analysis proved that the one-factor model was adequate [51]. This instrument has not been validated for the Portuguese population, so, its measurement will be carried out in this study.

##### Online Harassment Scale

Lewis et al. [52] conceived nine items to assess online harassment using a seven-point Likert-type scale measuring the frequency of harassment (from 1—‘never’ to 7—‘all the time’) personally experienced by the respondent in the course of their work as journalists. The first item measured general online harassment, with the subsequent eight items measuring particular forms of it. However, in this study, it will only be used 8 items; the first item was excluded as its content is specifically aimed at journalists and not the general public. The authors do not report psychometric properties of the instrument. Again, this instrument has not been validated for the Portuguese population, so, its measurement will be carried out in this study.

##### Flaming Behaviors Scale

The items used to measure flaming behavior were based on scales developed by Turnage [7]. Aggression, intimidation, insults, uninhibited language, and sarcasm are characteristics of flaming behaviors [7]. Each item was rated on a seven-point Likert scale from 1—‘strongly disagree’ to 7—‘strongly agree’. The author [7] found a Cronbach alpha value of 0.87. Hwang et al. [53] also found a value of 0.87. This instrument has not been validated for the Portuguese population, so, its measurement will be carried out in this study.

##### Trolling Behaviors Scale

Howard et al. [54] developed the 3-item scale that measures the extent to which the participant debated with others online and had intentions to aggravate/irritate others online. These items were created by a focus group. “The items asking, ‘To what extent do you enjoy the following: Debating various topics with the intention to irritate/upset others’ and ‘To what extent do you enjoy the following: ‘Trolling’ on public forums’ were measured on a 5-point Likert scale ranging from 1—‘not at all’ to 5—‘very much’. The item asking, ‘Please indicate how much you agree with the following statement: I like to post memes and comments with the intent to aggravate or annoy others’ was measured on a 5-point Likert scale ranging from 1—strongly disagree to 5—strongly agree” [54] (p. 310). The authors do not report the value of the Cronbach alpha. This instrument has not been validated for the Portuguese population, so, its measurement will be carried out in this study.

### 2.3. Data Analysis

Data analysis includes procedures related to descriptive statistics: for continuous data, measures of centrality (mean), dispersion (standard deviation and range) and shape (skewness and kurtosis) were used; frequencies and percentages are also displayed. The normality of the items was assessed by skewness (SI < 3) and kurtosis (KI < 10) indexes suggesting non-severe violations of normality [55], utilizing SEM, and multicollinearity was assessed by tolerance (>0.100) and variance inflation factor (VIF) (<10) [56]. Pearson correlations were established for continuous variables and Spearman correlations when at least one of the variables was ordinal or nominal. Correlations between 0 and 0.3 are weak, between 0.3 and 0.5 are moderate, between 0.5 and 0.7 are strong and between 0.7 and 1 are very strong either positive or negative [57].

A Confirmatory Factor Analysis (CFA) with Maximum Likelihood Estimation (MLE) was carried out to confirm the models proposed by the authors of the original versions of the instruments. The model fit evaluation was based on Kline’s [58] thresholds concerning statistics tests and approximate fit indexes. A statistically non-significant model chi-square statistic, χ^2^, shows that the model fits the data acceptably; the higher the probability related to χ^2^, the closer the fit to the perfect one. A value of the parsimony-corrected index Steiger–Lind root mean square error of approximation (RMSEA) close to 0 and non-significant at the 0.05 level represents a good fit. Values of incremental fit index (IFI), Tucker–Lewis index (TLI) and the Bentler incremental comparative fit index (CFI), close to 1 (0.95 or better), are indicators of best fit, as well as goodness of fit index (GFI). Re-specification of the models allows to analyzing path estimates, standardized residuals of items, and the modification indices. Regarding construct validity, items with low individual standardized factor loadings (regression weights below 0.50) may eventually be removal. The modifications indices (MI) provide information about potential cross-loadings and error term correlations (modifications theoretically meaningful and MI > 11 were taken into account). Concerning the parsimony-adjusted index Akaike Information Criterion (AIC) and the standardized root mean square residual (SRMR) (over 0.10 suggests fit problems), the smallest the values the most parsimonious is the model, they allow to compare the fit across models.

Concerning Cyberstalking scale, it was not possible to confirm the model proposed by the authors through the CFA. Therefore, it was decided to carry out an exploratory factor analysis (EFA) with maximum likelihood factoring. Parallel analysis—principal components analysis with raw data permutation [59]—was performed to determine the number of components to extract. Extracted factors were rotated by varimax rotation and the number of factors was decided in consideration of the scree-plot, cumulative variance explained, interpretability, and Kaiser’s criterion. Before, the Kaiser-Meyer-Olkin (KMO) test (measures sampling adequacy) and Bartlett’s test of sphericity were conducted to evaluate the factorability. Significance of Bartlett’s test of sphericity should be less than 0.001, meaning that EFA can be applied to the obtained data.

To assess the model reliability, convergent and discriminant validity, Cronbach’s alpha coefficients, composite reliability (CR, 0.70 or higher suggests good model reliability), average variance extracted (AVE, 0.50 or higher suggests adequate convergence) and square root of AVE (higher than the highest correlation with any other latent variable) were used; if AVE is less than 0.50 and CR higher than 0.60, the convergent validity of the model is adequate [60].

To compare the means of two groups, the independent means *t*-test (assumes the normality and homoscedasticity of the distribution variable) was applied. To compare the means of more than two groups, the F-test was applied (it assumes that the variable is normally and independently distributed, with equal variances among groups). Chi-squared test compares the distribution of categorical variables. Three measures of the effect-size, Cohen’s d, phi and eta squared were used accordingly to the level of measurement of the variables; interpretation followed Cohen et al. [57] guidelines. Hierarchical regression analysis was applied to examine the personality predictors of specific online behaviors while controlling for gender and age. To assess mediation effects of online behavior between personality traits and addiction to Internet, a multiple mediation model was tested with dark personality traits as independent variables, specific online behaviors as mediators, addiction to Internet as the outcome variable, and gender and age as control variables. A moderation analysis was carried out. The statistical significance level was set at 0.05. Statistical analysis was performed using SPSS version 28 and AMOS version 28 (Armonk, NY: IBM Corp, 2021).

## 3. Results

### 3.1. Descriptives

The sample consists of 773 participants, of which 467 (60.4%) are women; the sample presents a mean age of 27.39 (*SD* = 11.93; ranging from 19 to 78 years). Most of the sample is in a romantic relationship (*N* = 442; 57.2%). The mean age of the years of educative is 13.06 (*SD* = 2.47).

Most of the sample spends between two and six hours a day on the Internet, in non-academic or professional tasks. The main activities carried out on the Internet are social networking, communicate and relate to others, listen to music online and search for information and news in general. Less common activities on the Internet are betting sites and looking for new friends (Table 1).

In Appendix B, there is a description of all items of all instruments used in this study. Regarding the Internet Addiction Test, item 1 (Do you find that you stay online longer than you intended?) is the one with the highest mean value and item 3 (Do you prefer the excitement of the Internet to intimacy with your partner?) the lowest. Concerning Dirty Dozen Dark Triad, item 1 (I tend to want others to pay attention to me.) is the one with the highest mean value and items 4 and 6 (I tend to exploit others towards my own end; I tend to not be too concerned with morality or the morality of my actions.) the lowest. With regard to the Cyberstalking Scale, item 10 (When you’re interested in someone, it’s not wrong to look at their acquaintances’ social media, in order to get to know them better.) is the one with the highest mean value and item 6 (If I had my partner’s social media passwords, my life would be easier.) the lowest. In what it refers to Online Harassment Scale, item 6 (Hurt emotionally or psychologically.) is the one with the highest mean value and item 4 (Threats of physical sexual violence.) the lowest. Regarding Flaming Behaviors Scale, item 5 (In the community, I tend to make sarcastic remarks about others opinion. (Sarcasm)) is the one with the highest mean value and item 2 (In the community, I tend to intimidate people who get on my nerves. (Intimidation)) the lowest. At last, concerning Trolling Behaviors Scale, item 1 (Debating various topics with the intention to irritate/upset others.) is the one with the highest mean value and item 2 (Trolling’ on public forums) the lowest (Appendix B).

Based on the normative values of skewness and kurtosis, utilizing SEM, proposed by Brown [55], data is considered to have a normal distribution if skewness is between −3 and +3 and kurtosis between −10 and +10. Item 4 of the Online Harassment Scale is slightly above the recommended values for skewness and well above recommended values for kurtosis (Appendix B). The literature suggests several options for dealing with skewness and kurtosis, including (1) doing nothing (because many statistical tests, such as *t* tests, ANOVAs, and linear regressions, aren’t very sensitive to skewed data, especially if the skew is mild or moderate); (2) use another model (such as non-parametric tests or generalized linear models); and (3) transform the variable if it becomes less skewed [61]. At this point of the analysis, it was decided not to make any changes to the data, waiting to see how these items behave in subsequent analyses.

### 3.2. Study 1

#### 3.2.1. Specific Objective (1) to Evaluate the Psychometric Qualities of the Instruments Used in This Study through (1a) Assessment of the Adjustment to Our Sample of Instruments Previously Validated for the Portuguese Population (Internet Addiction Test and Dirty Dozen Dark Triad)

The Internet Addiction Test was validated for the Portuguese population as being unifactorial. We proceeded to carry out a confirmatory factorial analysis to evaluate the model in our population, but the fit was unacceptable. After consulting the modification indices, two correlations between errors were established until a good model was found [χ^2^(149) = 3.30; IFI = 0.938; TLI = 0.921; CFI = 0.938; GFI = 0.936; SRMR = 0.046; RMSEA = 0.055 (LO90 = 0.049; HI90 = 0.060); AIC = 612.98]. Cronbach’s alpha coefficients (0.90) composite reliability (0.91), average variance extracted (AVE, 0.55) and square root of AVE (0.74) were calculated, being the values within the reference range. Mean (1.78) and standard deviation (0.64) were also calculated.

The Dirty Dozen Dark Triad was validated for the Portuguese population as tri-factorial. We proceeded to carry out a confirmatory factorial analysis to evaluate the model in our sample, but the fit was not good. After consulting the modification indices, two correlations between errors were established until a good model was found [χ^2^(41) = 3.66; IFI = 0.961; TLI = 0.937; CFI = 0.961; GFI = 0.970; SRMR = 0.046; RMSEA = 0.059 (LO90 = 0.049; HI90 = 0.069); AIC = 223.87]. Cronbach’s alpha coefficients (Machiavellianism α = 0.73; Psychopathy α = 0.68; Narcissism α = 0.81); composite reliability (Machiavellianism CR = 0.83; Psychopathy CR = 0.81; Narcissism CR = 0.88); average variance extracted (AVE) (Machiavellianism AVE = 0.56; Psychopathy AVE = 0.51; Narcissism AVE = 0.64); square root of AVE (Machiavellianism AVE^sr^ = 0.75; Psychopathy AVE^sr^ = 0.71; Narcissism AVE^sr^ = 0.80); mean and standard deviation were calculated (Machiavellianism *M ± SD* = 2.27 ± 0.81; Psychopathy *M ± SD* = 1.81 ± 0.73; Narcissism *M ± SD* = 2.38 ± 0.92). All the values were within the reference range.

#### 3.2.2. Specific Objective (1) to Evaluate the Psychometric Qualities of the Instruments Used in this Study through (1b) the Validation for the Portuguese Population of the Cyberstalking Scale, Online Harassment Scale, Flaming Scale and Trolling Scale

##### Cyberstalking Scale

A confirmatory factor analysis (CFA) was carried out (*N* = 773) in order to confirm the unidimensional model proposed by the original authors. However, the model was very poor [χ^2^(35) = 17.45; IFI = 0.783; TLI = 0.720; CFI = 0.782; GFI = 0.841; SRMR = 0.083; RMSEA = 0.147 (0.147-0.157); AIC = 657.39]; the estimated values of standardized regression weights are within the reference values (between 0.44 and 0.70). Cronbach’s alpha of the total is 0.84 and no item, if removed, increases Cronbach’s alpha. After consulting the modification indices, several correlations between errors were established but, in spite of that, a good model was not found. The authors decided to perform an Exploratory Factorial Analysis (EFA). To this end, the sample was randomly divided into two groups with almost the same number of participants: one group assigned to EFA (*N* = 387) and the other to subsequent CFA (*N* = 386). Factors were extracted by the maximum likelihood method and rotated by varimax rotation. The number of factors was decided in consideration of the parallel analysis, scree-plot, cumulative variance explained, interpretability, and Kaiser’s criterion. In the parallel analysis, 3 components had eigenvalues that were above the 95th percentile of eigenvalues of 1000 random datasets of the same dimension. The EFA resulted in a solution with ten items distributed by three factors (Table 2) that explains more than 60% of the variance, with good psychometric indicators. The structure obtained in EFA was confirmed by CFA, using the other half of the sample. While most of the indicators are within the benchmarks, some, notably, were not. We consulted the modification indices and found that they suggested three correlations between errors; once these correlations between errors were established (theoretically supported by the proximity of the content), the model turned out to be very good [χ^2^(31) = 2.54; IFI = 0.963; TLI = 0.946; CFI = 0.963; GFI = 0.959; SRMR = 0.083; RMSEA = 0.063 (0.046–0.081); AIC = 126.73]. Cronbach’s alpha (Table 2), composite reliability, average variance extracted (AVE) and AVE square roots of the Portuguese version of Cyberstalking Scale are within the recommended values (Table 3).

##### Online Harassment Scale

A confirmatory factor analysis (CFA) was carried out (*N* = 773) to confirm the unidimensional model proposed by the original authors. However, the model was not good [χ^2^(20) = 14.00; IFI = 0.875; TLI = 0.824; CFI = 0.874; GFI = 0.917; SRMR = 0.061; RMSEA = 0.130 (0.117–0.143); AIC = 312.02]; the estimated values of standardized regression weights are within the reference values (between 0.54 and 0.71). Cronbach’s alpha of the total is 0.83 and no item, if removed, increases Cronbach’s alpha. Also, in the parallel analysis, only 1 component had an eigenvalue above the 95th percentile of eigenvalues of 1000 random datasets of the same dimension. After consulting the modification indices, three correlations between errors were suggested and established and a very good model was found [χ^2^(17) = 4.11; IFI = 0.975; TLI = 0.958; CFI = 0.974; GFI = 0.977; SRMR = 0.030; RMSEA = 0.063 (0.048–0.079); AIC = 107.91]. Cronbach’s alpha coefficients (0.83) composite reliability (0.88), average variance extracted (AVE, 0.50) and square root of AVE (0.71) were calculated, being the values within the reference range. Mean (1.85) and standard deviation (0.87) were also calculated.

##### Flaming Behaviors Scale

A confirmatory factor analysis (CFA) was carried out (*N* = 773) to confirm the unidimensional model proposed by the original authors. However, the model was not good [χ^2^(5) = 17.77; IFI = 0.957; TLI = 0.913; CFI = 0.957; GFI = 0.953; SRMR = 0.038; RMSEA = 0.147 (0.121–0.145); AIC = 108.84]; the estimated values of standardized regression weights are within the reference values (between 0.65 and 0.83). Cronbach’s alpha of the total is 0.87 and no item, if removed, increases Cronbach’s alpha. Also, in the parallel analysis, only 1 component had an eigenvalue above the 95th percentile of eigenvalues of 1000 random datasets of the same dimension. After consulting the modification indices, two correlations between errors were suggested and established and a very good model was found [χ^2^(3) = 3.03; IFI = 0.997; TLI = 0.989; CFI = 0.997; GFI = 0.995; SRMR = 0.010; RMSEA = 0.051 (0.015–0.091); AIC = 33.10]. Cronbach’s alpha coefficients (0.87) composite reliability (0.91), average variance extracted (AVE, 0.67) and square root of AVE (0.82) were calculated, being the values within the reference range. Mean (2.10) and standard deviation (1.25) were also calculated.

##### Trolling Behaviors Scale

A confirmatory factor analysis (CFA) was carried out (*N* = 773) to confirm the unidimensional model proposed by the original authors. It was found a very good model [χ^2^(1) = 1.64; IFI = 0.999; TLI = 0.997; CFI = 0.999; GFI = 0.999; SRMR = 0.001; RMSEA = 0.029 (0.000–0.105); AIC = 11.64]; the estimated values of standardized regression weights are within the reference values (between 0.70 and 0.75). Cronbach’s alpha of the total is 0.76 and no item, if removed, increases Cronbach’s alpha. Also, in the parallel analysis, only 1 component had an eigenvalue above the 95th percentile of eigenvalues of 1000 random datasets of the same dimension. Composite reliability (0.86), average variance extracted (AVE, 0.68) and square root of AVE (0.82) were calculated, being the values within the reference range. Mean (1.56) and standard deviation (0.82) were also calculated.

#### 3.2.3. Correlations between Psychological Variables

All psychological variables are positively and significantly correlated ranging from 0.150 (between Dirty Dozen Psychopathy and Cyberstalking Time) and 0.535 (between Flaming and Trolling); besides, all variables have a normal distribution and absence of multicollinearity (Table 4).

### 3.3. Specific Objective (2) to Establish Associations between the Variables under Study and the Sociodemographic Variables

Age correlates negatively and significantly with almost all the psychological variables: Internet Addiction (*r* = −0.420; *p* < 0.001), Dirty Dozen Dark Triad Machiavellianism (*r* = −0.217; *p* < 0.001), psychopathy (*r* = −0.171; *p* < 0.001), narcisism (*r* = −0.116; *p* < 0.001), cyberstalking total (*r* = −0.176; *p* < 0.001), cyberstalking justification (*r* = −0.181; *p* < 0.001), cyberstalking time (*r* = −0.243; *p* < 0.001), online harassment (*r* = −0.354; *p* < 0.001), flaming behaviors (*r* = −0.322; *p* < 0.001), and trolling behaviors (*r* = −0.196; *p* < 0.001). Years of education correlate positively and significantly with Dirty Dozen Dark Triad narcisism (*r* = 0.103; *p* < 0.001) and with cyberstalking justification (*r* = −0.072; *p* = 0.045).

Machiavellianism, psychopathy and narcissism are significantly higher in men than in women. The same goes for flaming and trolling. Cyberstalking justification and time, as well as online harassment, show higher values in women than in men (Table 5). Also, those who are not in a romantic relationship show higher values in Internet addiction, psychopathy, cyberstalking time, online harassment and flaming (Table 6).

### 3.4. Study 2

#### Specific Objective (3) to Look for Relationships between Personality Traits and Online Behaviors

Hierarchical regression analyses were applied to examine the personality predictors of specific online behaviors while controlling for gender and age (Table 7). Being female is associated with cyberstalking total, cyberstalking justification, cyberstalking time and online harassment; being male is associated with flaming and trolling. Age is positively associated with cyberstalking control and negatively with all the other dimensions. Machiavellianism is positively associated with all the dimensions of this study. Psychopathy is positively associated with cyberstalking total, cyberstalking control, flaming and trolling. At last, narcissism is positively associated with all the dimensions, except online harassment and flaming.

### 3.5. Specific Objective (4) to Determine Whether Online Behaviors Moderate the Relation between the Dimensions of the Dark Triad of Personality and Internet Addiction

To achieve this moderation, an index of online behaviors (cyberstalking, online harassment, flaming and trolling) was developed. The results of the moderation (Figure 1) showed that online behaviors moderate the relation between the dimensions of the dark triad of personality and internet addiction [χ^2^(2) = 2.30; IFI = 0.997; TLI = 0.984; CFI = 0.997; GFI = 0.998; SRMR = 0.013; RMSEA = 0.041 (0.000–0.092); AIC = 30.60]. Also, Machiavellianism, Psychopathy and Narcissism were indirectly associated with addiction to Internet through online behaviors.

## 4. Discussion

The main objective of this study was to assess the moderation effects of online behaviors between personality traits and Internet addiction. Thus, several specific objectives were delimited. The first objective was to evaluate the psychometric qualities of the instruments used in this study by assessing the fit to our sample of instruments previously validated for the Portuguese population (Internet Addiction Test and Dirty Dozen Dark Triad), and the validation for the Portuguese population of the Cyberstalking scale, Online Harassment scale, Flaming scale and Trolling scale. The results reveal good psychometric properties for the four validated scales, which allowed us to continue with our objectives.

As for the Cyberstalking scale, in the present study the one-dimensional structure was not verified, and it was found three factors were named cyberstalking control (includes items that monitor the other’s online behavior); cyberstalking justification (includes items that justify or normalize the acts committed) and cyberstalking time (includes items that refer to the time spent online in the practice of cyberstalking). The alpha values found are close to the original scale. One of the advantages of this scale is that it considered not only romantic relationships (past, current, and desired), but also included persecution of acquaintances and people that the perpetrator suspects/dislikes [51].

Relatively to the Online Harassment scale model found, it is in line with the proposal of the original authors [52]. As for the Flaming Behaviors Scale, the alpha value obtained was identical to the original author Turnage [7] and the study by Hwang et al. [53]. Confirmatory factor analysis revealed good scale adjustment indices, allowing confirmation of the unifactorial model of the original instrument. In turn, the Trolling Behaviors Scale in the present sample proved to be a reliable measure to evaluate this online malicious behavior.

The adaptation of these instruments may constitute a contribution to research in the context of these online antisocial behaviors (cyberstalking, online harassment, flaming and trolling) in a Portuguese context; the results that individuals obtain in the analyzed behavior can be related to other variables that help to understand the complexity of it.

The second objective was to establish associations between the variables under study and the sociodemographic variables. The results show that age correlates negatively and significantly with most of the psychological variables, namely: Internet addiction, Dirty Dozen Dark Triad Machiavellianism, psychopathy, narcissism, cyberbullying total, cyberbullying justification, cyberbullying time, online harassment, flaming behaviors, and trolling behaviors. That is, as a person age they exhibit fewer problematic online behaviors. This can be explained due to exposure to technology from a very young age, it may be that older people have less technical competencies to use the Internet and especially social networks. In this sense, most investigations indicate that younger individuals tend to be more Internet addicts [1], with age being considered in the literature as a risk factor. Young people are more directed to the use of technology (e.g., communicating, playing), and are more likely to have problems related to its use unlike older people, who use it mainly for work purposes [62].

Concerning the years of education, they are positively and significantly correlated with the Dirty Dozen Dark Triad narcissism and with the justification of cyberbullying. With respect to the sex variable, Machiavellianism, psychopathy and narcissism are significantly higher in men than in women. The same is true for flaming and trolling. Cyberstalking justification and time, as well as online harassment show higher values in women than in men. These results contrast with other results that find similar values for men and women [9]. Although traditional stalking is mostly practiced by male perpetrators, women are more likely to engage in covert stalking behaviors such as cyberstalking and online harassment [51].

The third objective was to search for relationships between personality traits and online behaviors. The results of our study show that Machiavellianism is positively associated with all dimensions of the studied variables. Psychopathy is positively associated with cyberbullying total, cyberbullying control, flaming and trolling; and narcissism was positively associated with most dimensions, except with online harassment and flaming. These results are in agreement with other studies showing similar results [9,16,20]. For example, Machiavellianism has been associated with problematic network use, trolling with online games, and online self-promotion, which is consistent with the explanation given by other studies indicating that this correlation is expected, as Machiavellianism manipulative behavior can be shown to be problematic on the Internet, being violent and controlling [2,20,37].

For its part, narcissism is associated with online self-promotion behaviors and trying to be more popular, which can cause problematic Internet use [20,32,38,39]; this coincides with our results. Given that individuals with a high level of narcissistic trait tend to focus extensively on themselves; and to be characterized by the dominance, grandeur and devaluation of others, it is expected that they use the possibilities of social media to promote and exalt themselves [25]. This process can occur through the search for approval and admiration of others on social networks leading to the employment of excessive time on the Internet [32].

Psychopathy has been associated with cyberbullying, trolling, intimate partner cyberbullying, and violent gaming [9,20,32]. This is in line with our findings, which can be explained by emotional dysregulation and the search for greater sensations, as referred to above. The involvement of individuals with psychopathic traits in online antisocial behaviors can be facilitated by the characteristics of the Internet itself, namely anonymity and difficulty in identifying the aggressor [17].

And finally, the last objective was to determine whether dimensions of the dark triad of personality explain online behaviors and these last moderates the relation between dak traid and Internet addiction. Our results reveal that Machiavellianism, Psychopathy and Narcissism are associated with Internet addiction through online behaviors. These associations reinforce the behavioral characteristics of these individuals and how they may behave on the Internet, allowing us to suggest that dimensions of the dark triad of personality explain online behaviors [4,9,10,16,20,32,34,35,38,39,63]. Cyberstalking also requires excessive time to investigate public posts from different social media accounts and victim profiles [20]. This compulsive impulse to persecute others gives rise to excessive social media use, gaining the Internet great importance in the lives of individuals and tolerance for the time spent online, characteristic symptoms of addiction to the Internet [2,8].

## 5. Conclusions

The results of this study have theoretical and practical implications, since, on the one hand, they reinforces the findings of other studies showing that dimensions of the dark personality triad play an important role in Internet and social network addition [20,21], contributing to the literature; and, on the other hand, on a practical level, they allow to conduct awareness campaigns in communities, schools, and work to understand that one can be exposed to unpleasant situations due to behaviors that some people with personality traits of Machiavellianism, narcissism and/or psychopathy can cause problems affecting the mental, emotional and psychological health of others.

Despite these results, some limitations should be addressed, namely, the use of self-report instruments that may somehow influence the data obtained, being susceptible to arbitrariness of responses, subjectivity of participants in the perception of content and tendency to give socially desirable answers. Additionally, the cross-sectional nature of the study is highlighted, which leads to the impossibility of establishing cause and effect relationships between the variables.

Future studies may investigate the issue of Internet dependencies versus Internet dependency, to understand whether the addition, when occurs, is originated in relation to activities on the Internet (e.g., social networks, gambling, shopping) or if it is in relation to the Internet itself. Future investigations should continue to study the addition to the Internet and online antisocial behavior including specific samples of clinical/forensic context, involving individuals with diagnoses of personality disorders that constitute the Dark Triad.

## Figures and Tables

**Figure 1 ijerph-20-06136-f001:**
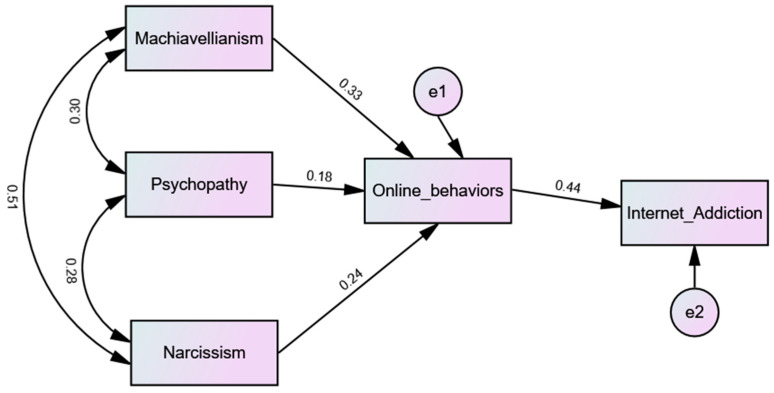
Final model.

**Table 1 ijerph-20-06136-t001:** Questionnaire about Internet use on mobile phone, computer or other devices (*N* = 773).

			*N*	%
How much time do you dedicate to the Internet per day, in non-academic or professional tasks?	1	Up to 2 h	185	23.9
	2	Between 2 to 4 h	253	32.7
	3	Between 4 to 6 h	234	30.3
	4	More than 6 h	101	13.1
Identify how you use the Internet (excluding academic or professional tasks).				
	No		Yes	
	** *N* **	**%**	** *N* **	**%**
1—Consult email	102	13.2	671	86.8
2—Social networks	23	3.0	750	97.0
3—Listen to music online	85	11.0	688	89.0
4—Looking for new friends	622	80.5	151	19.5
5—Search for information and news in general	98	12.7	675	87.3
6—Play online	475	61.4	298	38.6
7—Communicate and relate to others	81	10.5	692	89.5
8—Online shopping	263	34.0	510	66.0
9—Digress on the Internet	223	28.8	550	71.2
10—Watch movies/series	159	20.6	614	79.4
11—Betting sites	679	87.8	94	12.2
12—Search for adult content	598	77.4	175	22.6

*N* = frequencies; % = percentages.

**Table 2 ijerph-20-06136-t002:** Cyberstalking Scale: Exploratory Factorial Analysis.

	(*N* = 387) (10 Items)			
	*h* ^2^	*F1*	*F2*	*F3*
		Control	Justification	Time
1. I usually find the social media of someone I’m interested in, even if it takes hours.	0.704	0.034	0.280	0.791
2. It’s ok to check who likes and comments on the posts of your partner.	0.741	0.093	0.837	0.177
3. I lose track of time searching for information about my acquaintances on the Internet.	0.738	0.228	0.151	0.814
4. It is normal to “keep an eye” on the social media of someone who frequently interacts with your partner.	0.747	0.169	0.841	0.107
5. If a person hides their messages, I look for other ways to find out the content of them.	0.515	0.350	0.617	0.108
6. If I had my partner’s social media passwords, my life would be easier.	0.750	0.839	0.206	0.070
7. If I could I would look at my love partner’s browsing history.	0.711	0.791	0.291	0.019
8. I prefer to form relationships with people that I can investigate on social media.	0.702	0.702	−0.015	0.457
9. I check what kind of apps my partner uses on her/his phone.	0.562	0.678	0.282	0.153
10. When you’re interested in someone, it’s not wrong to look at their acquaintances’ social media, in order to get to know them better.	0.537	0.271	0.620	0.281
Eigenvalues		4.32	1.30	1.10
Total variance explained (67.07%)		43.23	12.98	10.87
Cronbach’s alfa (α)	0.84	0.81	0.79	0.65
Correlation matrix range [0.30–0.90]	0.19–0.69			
Determinant score [above 0.00001]	0.023			
Bartlett’s Test of Sphericity (*df*); *p* < 0.05	1441.77 (45); <0.001			
Kaiser-Meyer-Olkin Measure (KMO) (above 0.50)	0.846			
Diagonal element anti-correlation matrix (above 0.50)	0.81–0.91			

*h*^2^ = communalities; *F* = factor.

**Table 3 ijerph-20-06136-t003:** Correlations, Cronbach’s alpha, composite reliability, average variance extracted (AVE), AVE square roots, mean and standard deviation of the Dirty Dozen Dark Triad (*N* = 773).

	Pearson’s Correlations							
	1	2	3	4	α	CR	AVE	Mean (Standard Deviation)
1. Cyberstalking Total	**0.71**				0.84	0.88	0.50	1.93 (0.66)
2. Cyberstalking Control	0.80 **	**0.75**			0.81	0.84	0.57	1.48 (0.64)
3. Cyberstalking Justification	0.89 **	0.55 **	**0.73**		0.79	0.82	0.54	2.30 (0.89)
4. Cyberstalking Time	0.69 **	0.38 **	0.45 **	**0.80**	0.65	0.78	0.64	2.10 (0.97)

Note: ** *p* < 0.001; α = Cronbach’s alpha; CR = composite reliability; AVE = average variance extracted; **bold** (diagonal) = AVE square roots.

**Table 4 ijerph-20-06136-t004:** Pearson correlations between variables.

	1	2	3	4	5	6	7	8	9	10	11
1. Adiction Internet Total											
2. Dirty Dozen Machiavellianism	0.336 **										
3. Dirty Dozen Psychopathy	0.168 **	0.300 **									
4. Dirty Dozen Narcissism	0.254 **	0.510 **	0.278 **								
5. Cyberstalking Total	0.358 **	0.399 **	0.241 **	0.430 **							
6. Cyberstalking Control	0.234 **	0.307 **	0.254 **	0.327 **	0.803 **						
7. Cyberstalking Justification	0.290 **	0.343 **	0.181 **	0.383 **	0.892 **	0.554 **					
8. Cyberstalking Time	0.372 **	0.318 **	0.150 **	0.322 **	0.690 **	0.384 **	0.452 **				
9. Online Harassment Total	0.319 **	0.330 **	0.176 **	0.183 **	0.309 **	0.176 **	0.276 **	0.307 **			
10. Flaming Total	0.291 **	0.374 **	0.343 **	0.272 **	0.247 **	0.219 **	0.204 **	0.173 **	0.378 **		
11. TrollingTotal	0.221 **	0.310 **	0.278 **	0.247 **	0.224 **	0.188 **	0.176 **	0.188 **	0.278 **	0.535 **	
Skweness	0.212	0.188	0.983	0.212	0.640	1.424	0.457	0.688	1.547	1.404	1.744
Kurtosis	0.100	−0.715	0.632	−0.712	−0.053	1.827	−0.297	−0.354	2.918	1.658	2.650
Tolerance		0.626	0.813	0.662	0.877	0.639	0.587	0.713	0.761	0.606	0.686
VIF		1.598	1.230	1.511	1.140	1.564	1.704	1.402	1.314	1.649	1.457

Note: ** *p* < 0.001.

**Table 5 ijerph-20-06136-t005:** Differences in psychological variables concerning sex.

	Sex	*N*	*M*	*SD*	*t*(771)	*p*	*d*
1. Adiction Internet Total	Man	306	1.78	0.67	−0.11	0.910	0.64
	Woman	467	1.79	0.62			
2. Dirty Dozen Machiavellianism	Man	306	2.36	0.82	2.47	0.014	0.80
	Woman	467	2.22	0.80			
3. Dirty Dozen Psychopathy	Man	306	1.93	0.78	3.48	<0.001	0.72
	Woman	467	1.74	0.68			
4. Dirty Dozen Narcissism	Man	306	2.50	0.95	2.92	0.004	0.92
	Woman	467	2.30	0.90			
5. Cyberstalking Total	Man	306	1.85	0.62	−2.87	0.004	0.65
	Woman	467	1.98	0.68			
6. Cyberstalking Control	Man	306	1.51	0.63	0.99	0.322	0.64
	Woman	467	1.46	0.64			
7. Cyberstalking Justification	Man	306	2.15	0.79	−3.89	<0.001	0.89
	Woman	467	2.39	0.94			
8. Cyberstalking Time	Man	306	1.92	0.88	−4.11	<0.001	0.96
	Woman	467	2.21	1.01			
9. Online Harassment Total	Man	306	1.78	0.76	−1.97	0.049	0.87
	Woman	467	1.90	0.94			
10. Flaming Total	Man	306	2.48	1.42	6.66	<0.001	1.22
	Woman	467	1.85	1.06			
11. TrollingTotal	Man	306	1.78	0.93	5.70	<0.001	0.80
	Woman	467	1.42	0.70			

Note. *N* = frequencies; *M* = Mean; *SD* = Standard deviation; *t* = *t* test; *p* = significance; *d* = Cohen’s d.

**Table 6 ijerph-20-06136-t006:** Differences in psychological variables concerning romantic relationship.

	Romantic Relationship	*N*	*M*	*SD*	*t*(771)	*p*	*d*
1. Adiction Internet Total	No	331	1.97	0.63	7.19	<0.001	0.62
	Yes	442	1.65	0.61			
2. Dirty Dozen Machiavellianism	No	331	2.29	0.80	0.60	0.548	0.81
	Yes	442	2.26	0.81			
3 Dirty Dozen Psychopathy	No	331	1.89	0.77	2.41	0.016	0.73
	Yes	442	1.76	0.69			
4. Dirty Dozen Narcissism	No	331	2.38	0.93	0.04	0.969	0.93
	Yes	442	2.38	0.93			
5. Cyberstalking Total	No	331	1.95	0.61	0.89	0.371	0.66
	Yes	442	1.91	0.69			
6. Cyberstalking Control	No	331	1.48	0.62	0.03	0.975	0.64
	Yes	442	1.48	0.65			
7. Cyberstalking Justification	No	331	2.25	0.78	−1.30	0.193	0.89
	Yes	442	2.33	0.97			
8. Cyberstalking Time	No	331	2.31	0.98	5.35	<0.001	0.96
	Yes	442	1.94	0.93			
9. Online Harassment Total	No	331	1.94	0.85	2.63	0.009	0.87
	Yes	442	1.78	0.88			
10. Flaming Total	No	331	2.21	1.24	2.07	0.039	1.25
	Yes	442	2.02	1.26			
11. TrollingTotal	No	331	1.61	0.84	1.36	0.176	0.82
	Yes	442	1.53	0.81			

Note. *N* = frequencies; *M* = Mean; *SD* = Standard deviation; *t* = *t* test; *p* = significance; *d* = Cohen’s d.

**Table 7 ijerph-20-06136-t007:** Summary of the hierarchical regression analyses predicting different online behaviors.

		β (***t***)						
		Cyberstalking Total	Cyberstalking Control	Cyberstalking Justification	Cyberstalking Time	Online Harassment	Flaming	Trolling
Block 1	Gender	0.163 (5.222) ***	0.023 (0.679) ^ns^	0.185 (5.734) ***	0.182 (5.579) ***	0.095 (2.913) **	−0.200 (−6.491) ***	−0.169 (−5.119) ***
	Age	−0.075 (−2.353) *	0.075 (2.185) *	−0.098 (−2.974) **	−0.172 (−5.160) ***	−0.286 (−8.594) ***	−0.238 (−7.590) ***	−0.123 (−3.632) ***
Block 2	Machiavellianism	0.214 (5.766) ***	0.173 (4.347) ***	0.175 (4.585) ***	0.174 (4.486) ***	0.250 (6.472) ***	0.216 (5.901) ***	0.180 (4.587) ***
	Psychopathy	0.101 (3.049 **	0.161 (4.532) ***	0.055 (1.617) ^ns^	0.029 (0.832) ^ns^	0.060 (1.720) ^ns^	0.195 (5.976) ***	0.159 (4.538) ***
	Narcissism	0.301(8.287) ***	0.205 (5.250) ***	0.286 (7.630) ***	0.225 (5.907) ***	0.016 (0.422) ^ns^	0.059 (1.655) ^ns^	0.079 (2.048) *
	*R* ^2^ _adj_	0.264	0.154	0.217	0.195	0.198	0.285	0.174
	*F* _(3, 767)_	79.874	47.901	56.575	38.815	23.945	44.122	24.591
	*p*	<0.001	<0.001	<0.001	<0.001	<0.001	<0.001	<0.001

β = standardized Beta; *t* = *t* test; *** = *p* < 0.001; ** = *p* < 0.010; * = *p* < 0.050; ^ns^ = non significance.

## Data Availability

Data used in this study are available upon request.

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
