# Peer review of "Dark Personality Traits and Online Behaviors: Portuguese Versions of Cyberstalking, Online Harassment, Flaming and Trolling Scales"

_ijerph, 2023, doi:10.3390/ijerph20126136_

Round 1
Reviewer 1 Report
The manuscript submitted for publication covers a very important and current topic, which I read with great interest. The introduction to the literature is a carefully compiled and informative summary. However, I think that, in addition to the main objective (mediation effects of online behaviour between personality traits and addiction to Internet), the detailed psychometric analysis of the four scales is already too much for the length and clarity of the manuscript. I am aware that a reliable measurement instrument is an important part of exploring structural relationships, however, the psychometric analyses performed are already difficult to increase due to the limitations of the scope, however, I unfortunately see problems with the analysis of the scales. So, I would advise the authors to split this study into at least two papers for clarity.
My critical comments on the manuscript:
- Although the statistical analysis are thorough and well thought out, there are inaccuracies in several places: e.g., I find the following statement unsound and I could not find any reference to it in Kline's book: „Based on the normative values of skewness and kurtosis proposed by Kline [55], data is considered to have a normal distribution if skewness is between -3 and +3 and kurtosis between -10 and +10." I also feel that the lack of parallel analysis in determining the number of factors is a problem. Since the analyses in the manuscript are of good quality, I think it would be worthwhile to use this and additional indicators to support the one-dimensional structure. I feel that allowing correlations of errors is somewhat arbitrary and should not be used without an explanation of the overlap in the content of the items, just to improve the fit. Just a detailed psychometric analysis of the four scales would be worth some papers.
- After the EFA and CFA analyses, instead of the complicated path analysis model shown in Figure 5, it would be more appropriate to set up an SEM model (with fit indicators, possibly with latent variables). However, it would be worthwhile to move from simple to complicated in the model building. I feel that now a simple mediation analysis would be more impressive than the complex path analysis model outlined. It is raised that latent variables could be created from the different online behaviours, which could possibly make the results more transparent.
Overall, I find the manuscript very valuable, and I congratulate the authors. However, I am afraid that in the present format the key messages are lost and difficult to follow, so I suggest simplifying the manuscript to improve the clarity of the results.
Author Response
Dear Reviewer 1,
The authors' responses to the reviewer are below in italics.
The manuscript submitted for publication covers a very important and current topic, which I read with great interest. The introduction to the literature is a carefully compiled and informative summary.
The authors thank you for your words and appreciate your acknowledgment.
However, I think that, in addition to the main objective (mediation effects of online behaviour between personality traits and addiction to Internet), the detailed psychometric analysis of the four scales is already too much for the length and clarity of the manuscript.
The authors understand and agree to reduce the analysis, especially of the scales already validated for the Portuguese population.
I am aware that a reliable measurement instrument is an important part of exploring structural relationships, however, the psychometric analyses performed are already difficult to increase due to the limitations of the scope, however, I unfortunately see problems with the analysis of the scales. So, I would advise the authors to split this study into at least two papers for clarity.
My critical comments on the manuscript:
The authors propose to reduce the size of the document and at the same time divide the article into two studies (and not into two papers) with a view to clarifying them.
- Although the statistical analysis are thorough and well thought out, there are inaccuracies in several places: e.g., I find the following statement unsound and I could not find any reference to it in Kline's book: „Based on the normative values of skewness and kurtosis proposed by Kline [55], data is considered to have a normal distribution if skewness is between -3 and +3 and kurtosis between -10 and +10."
You are right – the authors corrected the author of the reference values.In yellow in text.
I also feel that the lack of parallel analysis in determining the number of factors is a problem.
Parallel analysis were performed for all four scales to be validated.
Since the analyses in the manuscript are of good quality, I think it would be worthwhile to use this and additional indicators to support the one-dimensional structure. I feel that allowing correlations of errors is somewhat arbitrary and should not be used without an explanation of the overlap in the content of the items, just to improve the fit. Just a detailed psychometric analysis of the four scales would be worth some papers.
The authors agree with the reviewer. However, it is increasingly difficult to publish articles with only instrument validations. In addition, in good Psychology journals, numerous scales with minimal validation are used. It seems to be a trend.
- After the EFA and CFA analyses, instead of the complicated path analysis model shown in Figure 5, it would be more appropriate to set up an SEM model (with fit indicators, possibly with latent variables). However, it would be worthwhile to move from simple to complicated in the model building. I feel that now a simple mediation analysis would be more impressive than the complex path analysis model outlined. It is raised that latent variables could be created from the different online behaviours, which could possibly make the results more transparent.
The authors agree with the reviwer and didi t – not a mediation but a moderation with a latent factor (online behaviors).
Overall, I find the manuscript very valuable, and I congratulate the authors. However, I am afraid that in the present format the key messages are lost and difficult to follow, so I suggest simplifying the manuscript to improve the clarity of the results.
The authors appreciate this comment.

Reviewer 2 Report
This study sought to identify the relationship between socially undesirable personality traits and addiction to the internet, considering several types of dysfunctional online behaviors. The authors conducted a thorough literature review and were meticulous in their study. The methods and results sections addressed all necessary steps for the proposed model, and the adaptations to Portuguese followed appropriate guidelines. I have only minor comments regarding the manuscript and would like to congratulate the authors on a well-done paper.
- In lines 26, 27, and 28, please indicate whether the association is positive or negative, similar to the sentences above.
- I advocate for providing detailed information at every step of the research. For example, what do the authors mean when they say at the end of the abstract, "theoretical and practical implications are discussed"? I suggest including one or two sentences summarizing the discussed implications.
- To ensure a broader readership for the manuscript, I suggest using keywords that do not repeat the information already stated in the title.
- In line 100, the reference used to justify the authors' argument regarding the inclusion of sadism discusses the adaptation of a Dark Triad measure but does not specifically address how sadism relates to the dark variables. I recommend the authors include more relevant references.
I suggest the authors consult the article below, especially because it has a full section discussing how Dark Tetrad traits are associated with online aversive behavior. Please feel free to include it or not.
Bonfá-Araujo, B., Lima-Costa, A. R., Hauck-Filho, N., & Jonason, P. K. (2022). Considering sadism in the shadow of the Dark Triad traits: A meta-analytic review of the Dark Tetrad. Personality and Individual Differences, 197, 111767."
- I strongly suggest the authors review sentence structure throughout the manuscript, as some information reads incorrectly. For example, "The mean age of the years of educative is 13.06 (SD = 2.47)."
Author Response
Dear Reviewer 2,
The authors' responses to the reviewer are below in italics.
This study sought to identify the relationship between socially undesirable personality traits and addiction to the internet, considering several types of dysfunctional online behaviors. The authors conducted a thorough literature review and were meticulous in their study. The methods and results sections addressed all necessary steps for the proposed model, and the adaptations to Portuguese followed appropriate guidelines. I have only minor comments regarding the manuscript and would like to congratulate the authors on a well-done paper.
The authors thank you for your words and appreciate your acknowledgment. All suggested changes have been made and are in yellow in the text.
- In lines 26, 27, and 28, please indicate whether the association is positive or negative, similar to the sentences above. Done
- I advocate for providing detailed information at every step of the research. For example, what do the authors mean when they say at the end of the abstract, "theoretical and practical implications are discussed"? I suggest including one or two sentences summarizing the discussed implications. Done
- To ensure a broader readership for the manuscript, I suggest using keywords that do not repeat the information already stated in the title. Done
- In line 100, the reference used to justify the authors' argument regarding the inclusion of sadism discusses the adaptation of a Dark Triad measure but does not specifically address how sadism relates to the dark variables. I recommend the authors include more relevant references. Done
I suggest the authors consult the article below, especially because it has a full section discussing how Dark Tetrad traits are associated with online aversive behavior. Please feel free to include it or not. Bonfá-Araujo, B., Lima-Costa, A. R., Hauck-Filho, N., & Jonason, P. K. (2022). Considering sadism in the shadow of the Dark Triad traits: A meta-analytic review of the Dark Tetrad. Personality and Individual Differences, 197, 111767." Done

Round 2
Reviewer 1 Report
Thanks for the corrections and congratulations to the authors for an excellent paper.